# Mechanical Changes of the Lumbar Intervertebral Space and Lordotic Angle Caused by Posterior-to-Anterior Traction Using a Spinal Thermal Massage Device in Healthy People

**DOI:** 10.3390/healthcare9070900

**Published:** 2021-07-15

**Authors:** Yong-Soon Yoon, Jong-Hoo Lee, Mihyun Lee, Ka-Eun Kim, Hong-Young Jang, Kyu-Jae Lee, Johny Bajgai, Cheol-Su Kim, Il-Young Cho

**Affiliations:** 1Presbyterian Medical Center, Department of Physical Medicine & Rehabilitation, 365, Seowon-ro, Wansan-gu, Jeonju-si 54987, Korea; gvcdr@daum.net (Y.-S.Y.); 2jhoo@naver.com (J.-H.L.); 2Department of Physical Education, Sungkyul University, 53, Seonggyeoldaehak-ro, Manan-gu, Anyang-si 14097, Korea; ksme_1998@naver.com; 3College of Medical Sciences, Jeonju University, 303, Cheonjam-ro, Wansan-gu, Jeonju-si 55069, Korea; kecam07@jj.ac.kr; 4Department of Medical Sciences Convergence Research Center for Medical Sciences, Jeonju University, 303, Cheonjam-ro, Wansan-gu, Jeonju-si 55069, Korea; brighthong0@jj.ac.kr; 5Department of Environmental Medical Biology, Wonju College of Medicine, Yonsei University, Wonju 26426, Korea; medbio9@gmail.com (K.-J.L.); johnybajgai@gmail.com (J.B.); cs-kim@yonsei.ac.kr (C.-S.K.)

**Keywords:** vertical traction, spinal thermal massage device, lumbar lordosis angle, intervertebral height, magnetic resonance imaging

## Abstract

Background: The axial (horizontal) traction approach has been traditionally used for treatment of low back pain-related spinal disorders such as nuclear protrusion, primary posterolateral root pain, and lower thoracic disc herniation; however, it is known to have some technical limitations due to reductions of the spinal curve. Lumbar lordosis plays a pivotal function in maintaining sagittal balance. Recently, vertical traction and combination traction have been attracting attention due to improving therapeutic outcomes, although evidence of their clinical application is rare; therefore, this study was conducted to investigate the mechanical changes of lumbar intervertebral space, lordotic angle, and the central spinal canal area through vertical traction treatment using a spinal massage device in healthy participants. Methods: In total, 10 healthy subjects with no musculoskeletal disorders and no physical activity restrictions participated. The participants lay on the experimental device (CGM MB-1901) in supine extended posture and vertical traction force was applied in a posterior-to-anterior direction on the L3–4 and L4–5 lumbar sections at level 1 (baseline) and level 9 (traction mode). Magnetic resonance (MR) images were recorded directly under traction mode using the MRI scanner. The height values of the intervertebral space (anterior, center, and posterior parts) and lordosis angle of the L3–4 and L4–5 sections were measured using Image J software and the central spinal canal area (L4–5) was observed through superimposition method using the MR images. All measurement and image analyses were conducted by 2 experienced radiologists under a single-blinded method. Results: The average height values of the intervertebral space under traction mode were significantly increased in both L3–4 and L4–5 sections compared to baseline, particularly in the anterior and central parts but not in the posterior part. Cobb’s angle also showed significant increases in both L3–4 and L4–5 sections compared to baseline (*p* < 0.001). The central spinal canal area showed a slightly expanded feature in traction mode. Conclusions: In this pilot experiment, posterior-to-anterior vertical traction on L3–4 and L4–5 sections using a spinal massage device caused positive and significant changes based on increases of the intervertebral space height, lumbar lordosis angle, and central spinal canal area compared to the baseline condition. Our results are expected to be useful as underlying data for the clinical application of vertical traction.

## 1. Introduction

Intervertebral discs (IVDs) are cushions of fibrocartilage and the principal joints between vertebral bodies in the spinal column. In addition, the lumbar lordosis curve plays a pivotal role in maintaining sagittal balance. IVDs are closely related to various spinal diseases including low back pain (LBP) because they function as a shock absorber inside the spine and reduce friction between vertebral bodies [1,2]. The damage or degeneration of IVD can lead to the mechanical compression of the spinal nerve or chemical stimulation, consequently causing various symptoms such as LBP, osteoarthrosis, loss of cartilage, and osteophyte formation in the facet joints and may lead to the development of spinal stenosis or degenerative spondylolisthesis [3]. To solve spinal dysfunction and LBP, many treatment measures focused on improving and modifying IVD physiology have been gaining attention, and related studies are advancing rapidly [1,2,4,5]. Studies have reported that the lifetime prevalence of LBP is as high as 84% and known to cause massive individual and economic burdens in industrialized nations such as Korea [6,7,8]. In addition, numerous studies have shown that adolescent LBP is a strong predictor for chronic LBP problems in adulthood [6,9,10,11]. Most cases of LBP are due to herniated discs, 98% of which occur in the L4–5 disc [2]. Lumbar lordosis, a key role in maintaining sagittal balance, is a notable factor affecting LBP [12]. It has been suggested that flattening and loss of the normal lumbar lordosis angle is an important clinical sign of back problems [12]. Almost 40% of overall lordosis is contributed by the last lumbar segment, L5, while only 5% is contributed by the L1 segment [13].

Spinal traction, a form of decompression therapy, traditionally has been used for the treatment of IVD-related diseases such as herniated discs, sciatica, degenerative disc disease, pinched nerves, and back pain [4,14,15,16]. Traction treatment causes a disc suction effect via mechanical distraction between vertebral bodies and by lowering the intradiscal pressure [17,18]. In this context, Krause and colleagues described that lumbar traction restores the neurological deficit and reduces pain by expanding the intervertebral distance and mitigating direct pressure or contact with damaged nerve tissue [16]. DeLacerda and colleagues [19] reported that traction had pain relief effects through increased blood flow by reducing the pressure on the nerves and improving stiffness and adhesion of spinal structures.

Until now, the spinal traction devices used for patient treatment have mostly adopted the axial traction approach; however, this approach has technical limitations due to the changes of the natural lordotic curve [20,21]. When axial traction force is applied in the supine position, the straightened spinal structures decrease the lumbar lordotic angle, which may cause negative effects such as muscle pain or spasms, as well as structural damage to the posterior facet joint, posterior longitudinal, interspinous ligament, and soft tissue, rather than decompressing IVD [20,21]. To address this limitation, posterior-to-anterior vertical traction and combination traction (axial and vertical traction) are good solutions due to maintaining the lumbar lordosis curve in supine posture [22,23]. In the randomized controlled study by Lee and colleagues, vertical traction led to increases of IVD spaces and expansion of the central spinal canal by maintaining lordotic curves, while also markedly improving the lumbar lordotic angle [23]. In addition, in a study on combination-type traction treatment, Park and colleagues demonstrated that it had beneficial effects for reducing tensile stress on the fibers of the annulus fibrosus in the posterior region and posterior longitudinal ligaments, reducing intradiscal pressure [24].

Recently, our experimental device involving a vertical traction function was developed and approved as a medical device for the purpose of muscular pain relief by the Korean Ministry of Food and Drug Safety [22]. The massage rollers, which are equipped on the mat of this device, move horizontally along the spinal axis of the patient in supine extended posture and ergonomically lift particular portions of the cervix and lumbar segments in the posterior-to-anterior direction according to the programmed mode. The massage rollers move according to the spinal curve, stopping at particular portions of cervix and lumbar segments, which are then lifted by vertical traction force in the posterior-to-anterior direction. This vertical lifting provides an intermittent vertical traction effect and consequently causes distraction effects on intervertebral bodies and decompression effects on IVDs, maintaining the lumbar curve; however, in this study we applied only the traction force without thermal function or horizontal massage in order to exclude the factors that can affect the traction results. For this, we manufactured the experimental device using plastic and wood material considering the simultaneous progress of traction treatment and magnetic resonance imaging (MRI).

Despite the many benefits of spinal traction treatment, clinical studies on existing axial traction therapy for lumbar disc disease patients are insufficient. Moreover, the limitations of horizontal traction therapy have been shown in the clinical research outcomes, especially in terms of reduced lordotic curvature [20,21]. Recently, vertical traction in supine posture has received attention as a vulnerability complement to traditional traction therapy, although scientific data and evidence are scarce; therefore, we conducted this pilot study to obtain objective data for real-time vertical traction applications. We hypothesized that posterior-to-anterior vertical traction on lumbar segments using a modified traction device would increase the intervertebral space height and lordotic angle, which was confirmed through MRI analysis.

## 2. Materials and Methods

### 2.1. Participants

A total of ten healthy adult volunteers (female: 4, male: 6, age: 28.1 ± 8.9 years, height: 171 ± 10 cm, weight: 74.8 ± 20.7 kg, body mass index: 27.1 ± 5.5 kg/m^2^) were recruited from October 2020 to January 2021. These participants were selected based on the relevant inclusion and exclusion criteria, as shown in Table 1. Demographic data were recorded for all participants. All participants signed the informed consent prior to the study commencement and ethical approval was granted by the Institutional Research Ethics Review Board, Presbyterian Medical Center (IRBN.2020-09-045). All participants voluntarily participated in the study after receiving a sufficient explanation of the purpose and method of the study from the clinical investigator. The participants received two interventions depending on the traction site (L3–4 and L4–5), with a washing period of 1 week between each intervention.

### 2.2. Measurement Tools and Research Design

In this study, the experimental device was slightly modified for the clinical research setting. It was manufactured using plastic and wood materials so as not to affect the MRI scanner, although it was the same as the commercial spinal thermal massage device (CGM MB-1901, CERAGEM Co. Ltd., Cheonan, Korea) in terms of its size, shape, and operating principle. It was composed of a main bed and an auxiliary mat on which massage rollers were equipped. In this study, the thermal function and translation massage function were excluded because they may have caused interference with the protraction results. We investigated the mechanical changes in traction through MRI image analysis by using the modified spinal massage device on L3–4 and L4–5 sections. Initially, participants lay down in the supine position with legs extended on the auxiliary mat, with the massage roller located under the L3–4 portion (Figure 1). For the MRI scan, the auxiliary mat where the participant was lying was moved into the MRI machine and then image scanning of the participant was conducted directly under traction mode in the MRI scanner (3 Tesla MRI, Siemens Healthcare GmbH, Erlangen, Germany). MRI scanning was performed using T2-weighted images (field of view: 150 × 150 mm; echo time: 108 ms; repetition time: 4500 ms). For the baseline treatment (*n* = 10), level 1 traction force was applied to the L3–4 portion and MRI scanning was performed. After 30 min of level 1 treatment, level 9 traction was conducted in the same way. After a one week washing period, L4–5 traction treatment and MRI scanning were performed at levels 1 and 9 in the same way as for the L3–4 treatment process. The difference in height for the massage roller between level 1 and level 9 was 62 mm (Figure 2).

### 2.3. Measurements of Intervertebral Height and Lumbar Cobb’s Angle under Traction Mode

The height values for the lumbar intervertebral space (L3–4 and L4–5, respectively) were measured using MR images and image processing software (Image J, National Institutes of Health, USA). Posterior and anterior midpoints, center points, midlines, and their bisectors were constructed (Figure 3) [25]. The posterior intervertebral height was given by the sum of the perpendicular distance from corners 1 to 3 (h1 + h3) and of the perpendicular distance from corners 2 to 4 (h2 + h4) to the bisector between the two midlines. The central intervertebral height was calculated by the sum of the perpendicular distances from the midpoints of superior and inferior endplate lines to the bisector (h5 + h6). Lumbar lordosis under traction at L3–4 and L4–5 was defined as the Cobb’s angle subtended by the inferior endplate line of L1 and the superior endplate line of S1 (Figure 3). The evaluations were carried out by 2 experienced radiologists using a single-blinded method to reduce measurement error. Each measurement was repeated three times by the same person to increase the repeatability.

### 2.4. Morphological Observation of Spinal Structures in Axial View of L4–5 Disc

To assess the changes in the central lumbar canal area before and after lumbar traction, the axial MRIs at the L4–5 disc level were observed by 2 experienced radiologists using a single-blinded method. The central spinal canal area was observed by superimposition method of baseline and traction MR images. To visualize the structural changes, identical sagittal-plane images at baseline and after level 9 lumbar vertical traction were superimposed using Adobe Photoshop.

### 2.5. Statistics

Statistical analysis in this study was carried out using Scientific Package for Social Sciences (version. 22; SPSS, Chicago, IL, USA). Descriptive statistics were performed to calculate the mean and standard deviation for all related data. Additionally, in order to evaluate the mechanical changes of the intervertebral height and Cobb’s angle for L3–4 and L4–5 sections, respectively, a paired sample t-test was conducted compared to baseline treatment. The statistical significance level was set at *p* < 0.05.

## 3. Results

### 3.1. Changes in Height of the Intervertebral Space in L3–4 and L4–5 Sections between Baseline and Traction Mode

The height values of the intervertebral space were measured in 3 parts (anterior, center, and posterior) in L3–4 and L4–5 sections, then the average height was calculated (Table 2). Our results showed that the intervertebral height values for the anterior side were significantly increased under traction mode in L3–4 (12.97 ± 1.90 mm vs. 14.26 ± 1.42 mm; *p* < 0.001) and L4–5 (15.01 ± 2.5 mm vs. 15.77 ± 2.48 mm; *p* < 0.001) sections compared to the baseline. Likewise, the intervertebral height values for the center showed significant increments in L3–4 (12.66 ± 1.53 mm vs. 13.30 ± 1.14 mm; *p* < 0.001) and L4–5 (11.97 ± 1.77 mm vs. 12.67 ± 1.61 mm; *p* < 0.01) sections under traction mode; however, the intervertebral height values for the posterior side in both L3–4 and L4–5 showed slight increments in disc height but not significant differences (Table 2). Average height values for the intervertebral space were significantly increased under traction in L3–4 (11.48 ± 1.27 mm vs. 12.21 ± 1.03 mm; *p* < 0.001) and L4–5 (11.57 ± 1.60 mm vs. 12.21 ± 1.39 mm; *p* < 0.001) sections compared to the baseline.

### 3.2. Changes in Lordotic Curve in L3–4 and L4–5 Sections in Baseline and Traction Mode

Table 3 summarizes the changes in Cobb’s angle under traction as compared to the baseline. As a result, the lordotic angle shows significant increases in traction mode in L3–4 (53.48 ± 8.21° vs. 68.55 ± 4.60°; *p* < 0.001) and L4–5 (55.17 ± 10.38° vs.67.95 ± 5.44°; *p* < 0.001) sites compared to baseline.

### 3.3. Observation of MR Images of the Central Spinal Canal Area

Next, two single-blinded radiologists observed the morphology of the spinal structure using MR images. The results showed expansion of the spinal canal area after traction compared to baseline. All 10 participants’ MRI findings showed consistent results, with representative MR images of both groups shown in Figure 4.

## 4. Discussion

Our study was conducted to investigate the biomechanical changes of posterior-to-anterior traction on lumbar segments using a modified traction device in healthy people to ensure safety. In this study, we applied a new concept of vertical traction in the posterior-to-anterior direction in supine extended posture and analyzed the height values of the lumbar intervertebral space and lordotic angles in L3–4 and L4–5 sections through MRI scanning a in clinical setting.

Traction is a well-known technique in which pulling force is used to alleviate the LBP via nerve root decompression through stretching of soft tissues and distracting the intervertebral space [16,18]. Worldwide, this technique is recommended by many physicians for conditions such as protruded IVDs to treat spinal muscle spasm, pain, and stiffness [26,27]. To date, different types of traction methods such as skeletal traction, spinal decompression, and mechanical traction are commonly used, with devices such as manual, mechanical, inversion, positional, and motorized instruments; however, despite its long history of use and academic results, the clinical usefulness of traction method in lumbar disc problems has been poorly recorded until now [20,21,23]. When traditional axial traction is applied to the patient in the supine position, the posterior spinal structures including the facet joints and posterior longitudinal and interspinous ligaments are stretched more than the anterior spinal structures due to decreased lordotic curve, consequently causing pain [21,24]; therefore, to complement this shortcoming of axial traction, new traction techniques and guidelines are required to achieve optimal therapeutic outcomes. In our study, we introduced vertical traction methods using the spinal massage device (Master V4, CGM MB-1901, CERAGEM Co. Ltd., Cheonan, Korea). This method maintains the natural spinal curve during the traction and furthermore improves the spinal structure. The massage rollers equipped in this device operate with an intermittent vertical traction function in the posterior-to-anterior direction in the supine extended position by mechanically lifting the vertebral segments locally. Additionally, a program configured in the device ergonomically scans and memorizes the individual spinal curves before treatment, subsequently detecting the exact traction point.

With this mechanism, we investigated the mechanical effects of vertical traction using the spinal massage device in healthy adults. Generally, axial traction increases the posterior disc height more than the anterior disc height, although the average disc height is significantly increased [28]; however, our MRI results showed significant increases of the intervertebral space in anterior and central parts of both L3–4 and L4–5 sections after vertical traction compared to baseline, but not in posterior parts of the L3–4 and L4–5 sections. Consequentially, average height values of the intervertebral space were significantly increased in both L3–4 and L4–5 segments under traction. The increases in the anterior, central, and average height of the intervertebral space indicate expansion of the intervertebral space, which can cause decompression of IVDs and a consequent suction effect on the protruded disc. As a result, insignificant posterior height of the intervertebral space indicates that posterior spinal structures, including posterior ligaments, receive little tension due to maintaining the lordotic curve as compared to the baseline condition. Our results are consistent with the research findings from the study conducted by Lee and colleagues who demonstrated that the anterior/posterior intervertebral distance ratio was significantly greater during lordotic curve-controlled traction (LCCT) compared to axial traction treatment in the L3–4 and L4–5 sections [20]. These findings suggest that posterior-to-anterior vertical traction might contribute to symptom improvement of the disc herniation. Additionally, our results showed significant increased lordotic angles without any discomfort. Previously conducted studies have demonstrated that the curvature of the spine is essential in the diagnosis of pathological abnormalities of LBP [29,30]. The lumbar curvature has been investigated in numerous studies, commonly by using Cobb’s method to evaluate the lordotic curve [29,30]. In the current study, we found that vertical traction at level 9 significantly improved the Cobb’s angle values of the L3–4 and L4–5 lumbar segments compared to baseline. Furthermore, the central canal area of the lumbar spine was found to be wider as compared to baseline. These results are supported by the comparative study between LCCT and traditional traction conducted by Lee and colleagues [20]. Their findings demonstrated that LCCT traction therapy resulted in greater improvements in pain, function, and morphology of the central canal area of the spine compared to the traditional axial traction technique in patients with lumbar intervertebral disc disease. Moreover, they also showed that LCCT traction increased the lordotic angles of the intervertebral space (L2–L5) in the spine in participants with non-radicular LBP [23]. In line with this, several reports have shown that local vertical traction treatment to maintain the spinal curve has pain relief effects [31,32,33]. In the same context, our results imply that traction in the posterior-to-anterior direction in supine posture using a spinal massage device can be more effective compared to axial traction for patients with lumbar radicular pain caused by disc herniation and with low back pain caused by excessive stretching of the posterior muscles and ligaments.

This study was conducted to verify mechanical changes of the vertical traction using the newly developed spinal massage device through MR images. As a result, we found that the spinal massage device (CGM MB-1901) used in this experiment served the purpose of traction treatment. The vertical traction force of level 9 was enough to cause structural changes of lumbar segments on the basis of significantly increased intervertebral height, lordotic angle, and central canal area values, without excessive stretching of posterior spinal structures [34,35]. In this study, we excluded the thermal and translation massage functions from the device to verify only the vertical traction effect; however, considering the fact that this device has been approved already as a medical device for the purpose of muscular pain relief by the Korean Ministry of Food and Drug Safety, we expect more improved traction outcomes for this device [22].

We designed this study to obtain vertical traction data, to ensure the safety of the spinal thermal massage device, and due to the lack of previous studies; therefore, we believe that the findings of this study might serve as a useful reference for the clinical study of lumbar vertebral segments using vertical traction devices and MR images, although our study has several limitations. First, due to the smaller sample size, the study results are difficult to generalize. Second, as this study was conducted in healthy young participants for safety purposes, the results obtained from this study may not be applicable to subjects with LBP. As such, further randomized clinical trials considering wider ranges in participant age and with a particular spinal disease are needed. Third, we did not investigate the long-term treatment efficacy of this traction device with an appropriate control group; therefore, further studies are required to establish optimal guidelines for the clinical settings for applicable spinal diseases.

## 5. Conclusions

Based on the MRI findings, vertical traction in the posterior-to-anterior direction using the newly conceptualized traction device (CGM MB-1901) with supine extended posture resulted in significant biomechanical changes of lumbar intervertebral height, lordotic angle, and spinal central canal area in L3–4 and L4–5 sections compared to the baseline mode of the traction device in healthy people. The significantly increased intervertebral height values in anterior and central regions but not in posterior regions suggested that vertical traction treatment can compensate for the limitations of traditional (axial) traction therapy by maintaining the natural lumbar curve, without excessive tension on posterior spinal structures; however, further studies are required to optimize clinical treatment. Accordingly, this study may be useful for the establishment of a vertical traction therapy database.

## Figures and Tables

**Figure 1 healthcare-09-00900-f001:**
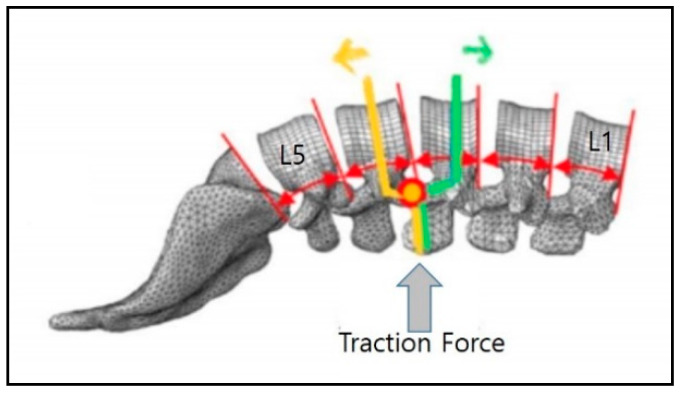
The traction position for vertical traction of the L3–4 section.

**Figure 2 healthcare-09-00900-f002:**
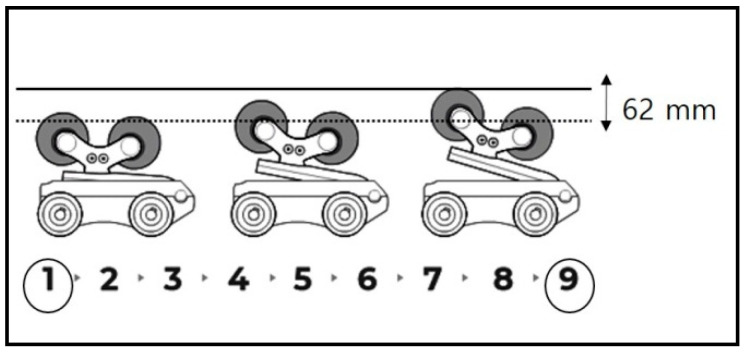
Difference in traction height for the massage roller between level 1 (baseline) and level 9 (traction mode).

**Figure 3 healthcare-09-00900-f003:**
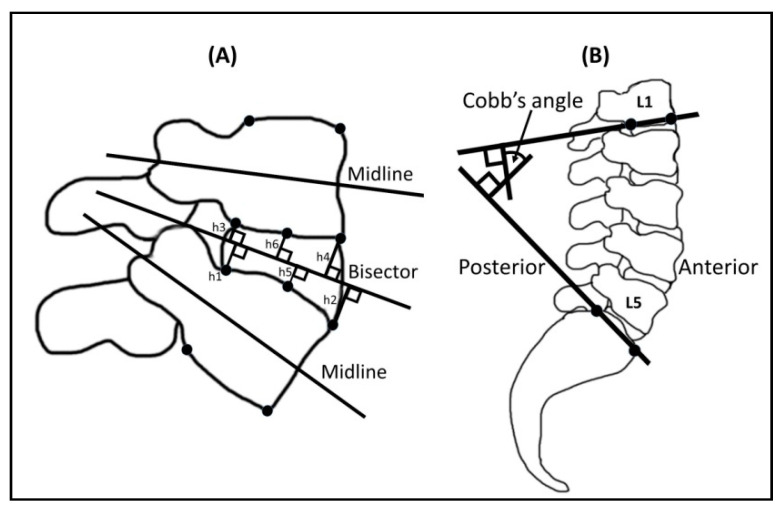
The measurements of intervertebral height (**A**) and lumbar lordosis angle (**B**). Posterior intervertebral height: h1 + h3; anterior intervertebral height: h2 + h4; central intervertebral height: h5 + h6.

**Figure 4 healthcare-09-00900-f004:**
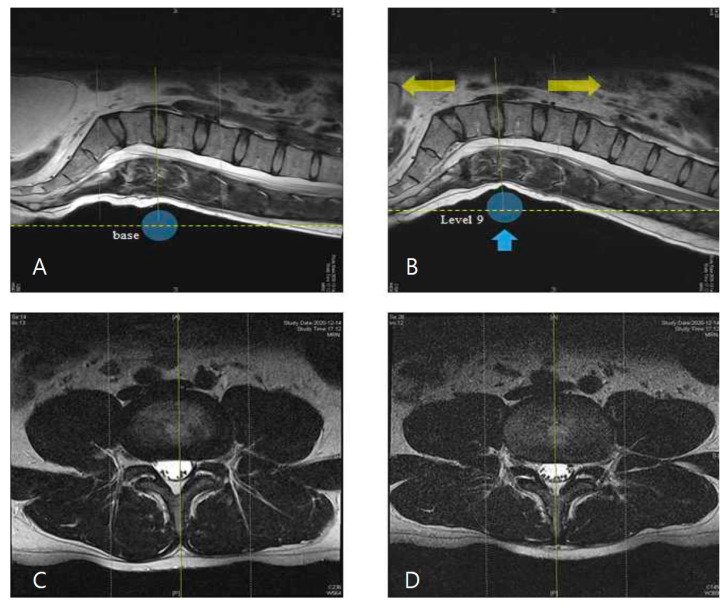
MR images of the central spinal canal under traction: level 1 (baseline) (**A**,**C**); level 9 (traction mode) (**B**,**D**). These images were superimposed after the image transparency process using Adobe Photoshop. Expansion of the central canal area was observed in the traction images (**D**) compared to the baseline images (**C**).

**Table 1 healthcare-09-00900-t001:** Inclusion and exclusion criteria for the participants.

Inclusion criteria	Age: 19–40 years History of not suffering from low back pain People who can easily visit hospital during study period
Exclusion criteria	Pain in spinal column Inserted with medical devices such as artificial pacemakers, drug pumps, and metal implants Cognitive dysfunction, sensory impairment, and spinal surgery patients Person who have received invasive spine treatment within 4 weeks before screening Spinal posterior arch defects such as spinal bifida, disc protrusion with spinal cord compression Osteoporosis patients Pregnant and lactating women MRI phobia participants

**Table 2 healthcare-09-00900-t002:** Changes in height of the intervertebral space in lumbar segments under baseline and traction modes.

Lumbar Segments	Measurement Location	Baseline (mm)	Traction Mode (mm)	*p*-Value
L3–4	Anterior	12.97 ± 1.90	14.26 ± 1.42	<0.001
	Central	12.66 ± 1.53	13.30 ± 1.14	<0.001
	Posterior	8.80 ± 1.64	9.06 ± 1.77	0.286
	Average	11.48 ± 1.27	12.21 ± 1.03	<0.001
L4–5	Anterior	15.01 ± 2.5	15.77 ± 2.48	<0.001
	Central	11.97 ± 1.77	12.67 ± 1.61	<0.01
	Posterior	7.72 ± 1.64	8.19 ± 1.12	0.120
	Average	11.57 ± 1.60	12.21 ± 1.39	<0.001

**Table 3 healthcare-09-00900-t003:** Changes in Cobb’s angle of lumber sections under traction mode.

Variable	Baseline (°)	Traction Mode (°)	*p*-Value
L3–4	53.48 ± 8.21	68.55 ± 4.60	<0.001
L4–5	55.17 ± 10.38	67.95 ± 5.44	<0.001

## Data Availability

The data presented in this study are available in this article.

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
