# Peer review of "Mechanical Changes of the Lumbar Intervertebral Space and Lordotic Angle Caused by Posterior-to-Anterior Traction Using a Spinal Thermal Massage Device in Healthy People"

_healthcare, 2021, doi:10.3390/healthcare9070900_

Round 1
Reviewer 1 Report
“The Mechanical Changes of Lumbar Intervertebral Space and Lordotic Angle by Posterior-to-anterior Traction Using Spinal Thermal Massage Device in Healthy People”
Overall strengths of the article:
This manuscript explores the mechanical changes of lumbar intervertebral space, lordotic angle, and central spinal canal area through vertical traction treatment using a spinal massage device. In this study, the authors have investigated the mechanical changes such as lumbar intervertebral space height, lordosis angle, and central spinal canal area under vertical traction to L3-4 and L4-5 in healthy people using MR images. Traction is a well-known technique in which pulling force is used for alleviation of low back pain by nerve root decompression through stretching soft tissues and distracting intervertebral space. The technique is recommended for treating spinal muscle spasms, pain, and stiffness. The authors conclude that the results are expected to be useful as underlying data for the clinical application of vertical traction. Overall this manuscript is very interesting but I noticed some critical issues that are in the comments and need to be addressed.
Specific comments on weaknesses:
Major Critical Comments:
- The overall hypothesis is missing.
- Sample size too small with no control.
- Why only normal healthy individuals only tested? That can question its therapeutic value.
- Why only <40 year age groups were only included? I think more severe problems exist in the older age groups?
- Does the test group include both males and females?
- The introduction needs major revision, some statements were presented without proper reference (citations) e.g. “The damage or degeneration of IVD can lead to the mechanical compression of a spinal nerve or chemical stimulation, consequently cause various symptoms such as paralysis or back pain”, “Currently LBP is commonly found in approximately 80% of adults above 50 years of age and can be also found in young population” and many others. I suggest revisiting the introduction and review the citations.
- Biochemical changes can also occur due to mechanical traction, why not discuss them?
- Any deleterious effect of treatment? What will happen after long-term treatment?
- I could not get what the authors are concluding from this study, conclusions not clearly presented.
Minor points:
- Figure 3; not very clear, text font size should be bigger and line width should be higher for a clear view.
- Text formatting needed.
- References need to be formatted carefully.
Author Response
Please see the attachment below.

Reviewer 2 Report
The manuscript is well written and the research has been well validated with superior quality of experimental study, however, the sample size of mere 10 is quite insufficient to draw rightful conclusions of the effectiveness.
What might the results depict should a subjects under study suffer mild back ailments? It is necessary to address this.
What is the significance of parameters like weight (for healthy individuals), age (eg: bone resorption) have on the current study?
Round 2
Reviewer 1 Report
Thanks to the authors for the revised manuscript, this manuscript has been significantly improved and the authors have clearly addressed all the quarries raised by the reviewer.